# RECURRENT NEURAL NETWORKS ARE UNIVERSAL FILTERS

## ABSTRACT

Recurrent neural networks (RNN) are powerful time series modeling tools in machine learning. It has been successfully applied in a variety of fields such as natural language processing (Mikolov et al. (2010), Graves et al. (2013), Du et al. (2015)), control (Fei & Lu (2017)) and traffic forecasting (Ma et al. (2015)), etc. In those application scenarios, RNN can be viewed as implicitly modelling a stochastic dynamic system. Another type of popular neural network, deep (feed-forward) neural network has also been successfully applied in different engineering disciplines, whose approximation capability has been well characterized by universal approximation theorem (Hornik et al. (1989), Park & Sandberg (1991), Lu et al. (2017)). However, the underlying approximation capability of RNN has not been fully understood *in a quantitative way*. In our paper, we consider a stochastic dynamic system with noisy observations and analyze the approximation capability of RNN in synthesizing the optimal state estimator, namely optimal filter. We unify the recurrent neural network into Bayesian filtering framework and show that recurrent neural network is a *universal approximator of optimal finite dimensional filters* under some mild conditions. That is to say, for any stochastic dynamic systems with noisy sequential observations that satisfy some mild conditions, we show that (informal)

$$\forall \epsilon > 0, \exists \text{ RNN-based filter}, \text{s.t.} \limsup_{k \to \infty} \left\| \hat{x}_{k|k} - \mathbb{E}[x_k|Y_k] \right\| < \epsilon,$$

where $\hat{x}_{k|k}$ is RNN-based filter's estimate of state $x_k$ at step $k$ conditioned on the observation history and $\mathbb{E}[x_k|Y_k]$ is the conditional mean of $x_k$, known as the optimal estimate of the state in minimum mean square error sense. As an interesting special case, the widely used Kalman filter (KF) can be *synthesized by RNN*.

## 1 INTRODUCTION

Recurrent neural network (RNN) is a certain type of neural networks characterized by hidden variables that memorize the history of input sequences, and it has been successfully applied and brought amazing results in many different disciplines including computer vision, natural language processing and optimal control, etc. (Mikolov et al. (2010), Graves et al. (2013), Du et al. (2015), Fei & Lu (2017), Ma et al. (2015)). Its huge empirical success in different engineering disciplines is grounded on the expressive power of RNN. However, how to understand the expressive power of RNN *in a quantitative way* is not fully understood. Even what RNN expresses is not totally clear. Another type of neural network, deep feed forward neural network has been well characterized as a *universal function approximator* (Hornik et al. (1989), Park & Sandberg (1991)). However, a similar way to characterize the expressive power of RNN is not obvious.

DNN is a mapping from a finite dimensional Euclidean space to another finite dimensional Euclidean space, that is to say it can be regarded as a vector-valued multivariate function. However, RNN is a mapping from a sequence space to another sequence space and the current output depends on both current input and the whole observation history. The input sequence, in principle, can be infinite. The function of RNN is capturing the relationship between input process and output process. For example, in machine translation, the input process (or the observation process) is sentence in one language and the output process (or the state process) is sentence in another language. And in many other RNN's application scenarios such as traffic forecast and optimal control, the input is a noisy

observation or measurement sequence and the output is an estimate sequence of a certain quantity, e.g., the traffic speed.. We observe that the function of RNN is quite similar to a *filter*.

In our paper, we propose to characterize the expressive power of RNN in *a quantitative way* from the perspective of filtering. We consider a discrete filtering system as 1.

$$\begin{cases} x_k = f(x_{k-1}) + g(x_{k-1})w_{k-1}, \\ y_k = h(x_k) + v_k, \end{cases} \tag{1}$$

where the state $x_k$ at time instant $k$ is an $n$-dimensional vector, $f$ is an $n$-dimensional vector-valued function, $g$ is an $n \times r$ matrix-valued function, $\{w_k, k = 0, 1, \cdots\}$ is an $r$-dimensional white Gaussian process and $w_k \sim \mathcal{N}(0, Q)$, where $Q$ is the covariance matrix of $w_k$. $y_k$ is the $m$-dimensional observation (measurement) process, $h$ is an $m$-dimensional vector-valued function, $\{v_k, k = 1, \cdots\}$ is an $m$-dimensional white Gaussian process and $v_k \sim \mathcal{N}(0, R)$, where $R$ is the covariance matrix of $v_k$. And we assume that $\{w_k, k = 0, 1, \cdots\}$, $\{v_k, k = 1, \cdots\}$ and the initial state $x_0$ are jointly independent. We use $Y_k$ to denote the sequence of observations up to time instant $k$, i.e.,

$$Y_k := \{y_1, \cdots, y_k\}. \tag{2}$$

Given the realization of the sequence of observations $Y_k$, the aim of filtering problem is to compute the optimal estimate of $x_k$ conditioned on $Y_k$.

Not surprisingly, recurrent neural network has been proposed to do filtering. James Ting Ho Lo showed that Recurrent Multi-layer Perceptron can be used to synthesize optimal filter (Lo (1994)). However, Lo's approach is based on simply copying and storing the whole observation history in the hidden variables and thus require the time horizon to be finite, which is fundamentally limited. Besides Lo's work, many efforts have been made to connect RNN and dynamical system. Wilson & Finkel (2009) implemented a neural network based Kalman Filter (KF) but did not provide theoretical analysis on the approximation error. Parlos et al. (2001) proposed an algorithmic approach to do nonlinear filtering using recurrent neural network architecture but did not provide theoretical gurantee. Schäfer & Zimmermann (2006) shows that recurrent neural network is a universal approximator of dynamical system. However, they only consider the deterministic system dynamics and do not analyze the filtering relationship between two stochastic processes. Around the same time our work was under review, a similar paper was also uploaded in arXiv by Lim et al. (2019), The results in the arXiv paper are similar to the June, 2018 graduation thesis of the first author of the current paper in the use of sufficient statistics, and emphasize on the design and implementation of the algorithm. Our current paper has been dramatically improved by introducing the error analysis and asymptotic convergence of the algorithm. And our work is inspired by the finite dimensional filter such as Beneš filter Beneš (1981).

Compared to the existing work, we make the following specific **contributions**:

- Motivated by the similarity between RNN and filter, we propose to use the ability to approximate optimal filter to characterize the expressive power of RNN. Unlike existing work on expressive power of RNN (Schäfer & Zimmermann (2006)) where only deterministic system is considered, we consider a stochastic dynamic system with noisy observations and analyze the capability of RNN to estimate unknown state.

- We unify RNN-based filter into Bayesian filtering framework. In this framework, the hidden variables of RNN are interpreted as statistics of the observation history. And the evolution of hidden variables are interpreted as the evolution of statistics.

- Based on the Bayesian filtering framework, we derive our main result: Recurrent Neural Networks (RNN) are *universal approximators* of a large class of optimal finite dimensional filters. That is to say, RNN estimator's asymptotic estimation error can be as close to minimum mean square error as desired. As an interesting special case, the widely used Kalman Filter can be synthesized by RNN. The consideration of asymptotic error differentiates us from existing work on expressive power of RNN (Schäfer & Zimmermann (2006)).

## 2 PRELIMINARY: THE BAYESIAN FRAMEWORK OF FILTERING

We first introduce the definition of minimum mean square error estimate.

**Definition 1** (**M**inimum **M**ean **S**quare **E**rror (MMSE) Estimate (Jazwinski (1970))). *Let $\hat{x}$ be an estimate of random variable $x$ and $L_{\text{MSE}} := (x - \hat{x})^T(x - \hat{x})$. The estimate that minimizes $\mathbb{E}[L_{\text{MSE}}]$ is called the minimum mean square error estimate.*

**Theorem 1** (Theorem 5.3 in Jazwinski (1970)). *Conditioned on the observation history $Y_k$, the minimum mean square error estimate of state $x_k$ is the conditional mean $\mathbb{E}[x_k|Y_k]$.*

*Proof.* Proof can be found in our appendix A.1. □

The Bayesian filtering consists of recursive prediction and update procedures (Jazwinski (1970)):

**Prediction Step** $p(x_{k-1}|Y_{k-1}) \to p(x_k|Y_{k-1})$: Given the posterior distribution $p(x_{k-1}|Y_{k-1})$ at instant $k - 1$, the prior distribution $p(x_k|Y_{k-1})$ of $x_k$ satisfies the Chapman-Kolmogorov equation:

$$p(x_k|Y_{k-1}) = \int p(x_k|x_{k-1})p(x_{k-1}|Y_{k-1})\, \mathrm{d}x_{k-1}; \tag{3}$$

**Updating Step** $p(x_k|Y_{k-1}) \to p(x_k|Y_k)$: Given prior distribution $p(x_k|Y_{k-1})$, when the observation $y_k$ at instant $k$ arrives, the posterior distribution $p(x_k|Y_k)$ at instant $k$ is given by equation 4,

$$p(x_k|Y_k) = \frac{p(y_k|x_k)p(x_k|Y_{k-1})}{\int p(y_k|x_k)p(x_k|Y_{k-1})\, \mathrm{d}x_k}. \tag{4}$$

Then we can get the MMSE estimate by simply doing integration:

$$\mathbb{E}[x_k|Y_k] = \int p(x_k|Y_k)x_k\, \mathrm{d}x_k. \tag{5}$$

To facilitate subsequent discussion, we make the following definition.

**Definition 2** (Sufficient Statistic (Beneš (1981))). *If the conditional distribution $p(x_k|Y_k)$ (or $p(x_k|Y_{k-1})$) can be fully determined by a vector-valued function $s_{k|k}$ (or $s_{k|k-1}$) of the observation sequence $Y_k$ (or $Y_{k-1}$), then we say $s_{k|k}$ (or $s_{k|k-1}$) is a* sufficient statistic *for $p(x_k|Y_k)$ (or $p(x_k|Y_{k-1})$).*[1]

Because sufficient statistic $s_{k|k}$ fully determines the posterior distribution $p(x_k|Y_k)$ and MMSE estimate $\mathbb{E}(x_k|Y_k)$ is a functional of $p(x_k|Y_k)$, there exists a function $\gamma$ that maps $s_{k|k}$ to $\mathbb{E}[x_k|Y_k]$,[2] i.e.,

$$\mathbb{E}[x_k|Y_k] = \gamma(s_{k|k}). \tag{6}$$

## 3 RNN BASED FILTER'S ARCHITECTURE

Motivated by the Bayesian framework of filtering, we propose the RNN based filter's architecture as shown in Fig. 1.

Our RNN based filter's architecture, Bayesian Filter Net (BFN), consists of three parts: prediction network, update network and estimation network. To mimic the prediction step in Bayesian filtering, we use the prediction network to map the posterior distribution representation vector to a prior distribution representation vector. To mimic the update step in Bayesian filtering, we then use an update network to update the prior state distribution representation vector and the observation to get the posterior state distribution representation vector. Finally, we use an estimation network to map the current posterior state distribution representation vector to current estimation $\hat{x}_k$. We will see in the subsequent discussions, the so-called representation vector or hidden variables indeed can be interpreted as statistics of the underlying conditional distribution.

---

[1] The sufficient statistic $s_{k|k}$ (or $s_{k|k-1}$) can be any vector-valued function of $Y_k$ (or $Y_{k-1}$) as long as it can fully determine $p(x_k|Y_k)$ (or $p(x_k|Y_{k-1})$). Taking linear system as an example, $s_{k|k}$ can be the vector composed of the conditional mean and covariance because these two quantities sufficiently determine the conditional density function of the state in linear filtering system. (See more details in our Appendix A.3.)

[2] For instance, as for the Kalman Filter in linear system, the sufficient statistic $s_{k|k}$ can be chosen to be the conditional mean and conditional covariance, thus $\gamma$ function can be the function that copy the conditional mean part of $s_{k|k}$.

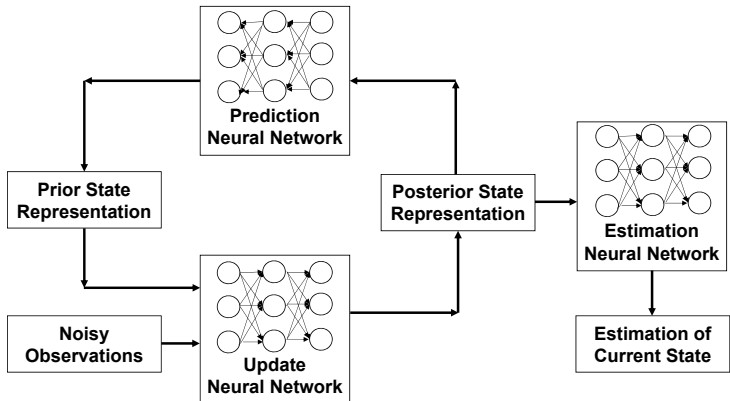

Figure 1: RNN based filter's architecture: Bayesian Filter Net (BFN) .

# 4    RNN BASED FILTER IS UNIVERSAL

We now show that our proposed RNN based filter is universal in that it can approximate a large class of optimal finite dimensional filters *to any asymptotic accuracy we desire*. We summarize our insight into the diagram 2.

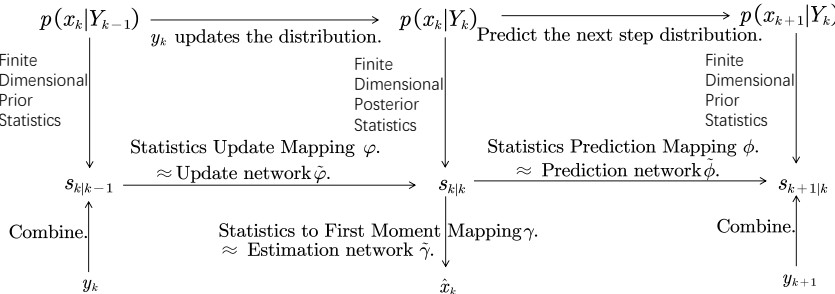

Figure 2: Our approach's illustrative diagram

As shown in the diagram 2, we model the conditional prior probability $p(x_k|Y_{k-1})$ $\big($posterior probability $p(x_k|Y_k)$ resp.$\big)$ of the state at step $k$ by finite dimensional prior statistics $s_{k|k-1}$ (posterior statistics $s_{k|k}$ resp.). In finite dimensional filter case (Beneš (1981; 1985); Daum (1987)), there exist finite dimensional statistics that are sufficient, that is to say, the evolution of conditional probability can be fully captured by the evolution of a finite dimensional vector. We denote the evolution function in updating step by $\varphi$ and the evolution function in prediction step by $\phi$. And further, after modelling the the probability distribution as finite dimensional statistics, we use two neural networks to approximate the evolution of them. And to get the final estimate of the state, we use another neural network to approximate the map from the statistics to the optimal estimation.

For the RNN based filter, one can naturally ask:

1. Will the neural network approximation error accumulate and blow up when time goes to infinity and make RNN asymptotically not work at all?

2. How general is this approach? What filter can RNN approximate?

We give the answers in this section.

### 4.1 KALMAN FILTER (KF) CAN BE SYNTHESIZED BY RNN

When system equation 1 is linear and satisfies Gaussian noise assumption as shown in equation 7, it is well known that the filtering problem can be optimally solved by Kalman Filter (KF) (Kalman (1960)). (See more details in our Appendix A.3.)

$$\begin{cases} x_k = F x_{k-1} + G w_{k-1} \\ y_k = H x_k + v_k, \end{cases} \tag{7}$$

where $F, G, H$ are constant matrices with proper dimensions, the initial state $x_0$ is Gaussian, and $\{w_k, k = 0, 1, \cdots\}$ and $\{v_k, k = 1, \cdots\}$ are two independent white Gaussian sequences that are also independent of the initial state $x_0$.

It can be known that the prior and posterior distributions are Gaussian and fully characterized by the sufficient statistics $s$ composed of mean and covariance matrix. (See more details in our Appendix A.3.)

$$\begin{cases} s_{k|k-1} := [m_{k|k-1}, \text{vec}^T(P_{k|k-1})]^T, \\ s_{k|k} := [m_{k|k}, \text{vec}^T(P_{k|k})]^T, \end{cases} \tag{8}$$

where $m_{k|k-1}$ and $P_{k|k-1}$ are the conditional mean and covariance in step $k$ conditioned on $Y_{k-1}$ and $m_{k|k}$ and $P_{k|k}$ are the mean and covariance in step $k$ conditioned on $Y_k$, and $\text{vec}(\circ_{n_1 \times n_2})$ is the $n_1 n_2 \times 1$ column vector obtained by stacking the columns of the matrix $\circ$ on top of one another. $s_{k|k}$ ($s_{k|k-1}$ resp.) is the theoretical statistic that determines the conditional probability distribution of the state $x_k$ conditioned on the observation history $Y_k$ ($Y_{k-1}$ resp.) and evolves according to some function $\varphi$ and $\phi$. (See more details in our Appendix A.3.) We also know that there exists some function $\gamma$ that maps $s_{k|k}$ to MMSE estimate $\mathbb{E}[x_k|Y_k]$. Thus we have,

$$s_{k|k} = \varphi(s_{k|k-1}, y_k), \ s_{k+1|k} = \phi(s_{k|k}), \ \mathbb{E}[x_k|Y_k] = \gamma(s_{k|k}). \tag{9}$$

We use function $\tilde{\varphi}$ generated by a deep (feedforward) neural network (i.e. update network) to approximate $\varphi$, and use $\tilde{\phi}$ generated by another DNN (i.e. prediction network) to approximate $\phi$. And the numerical statistics computed by RNN are denoted as $\tilde{s}_{k|k}$ and $\tilde{s}_{k+1|k}$, i.e.,

$$\tilde{s}_{k|k} = \tilde{\varphi}(\tilde{s}_{k|k-1}, y_k), \ \tilde{s}_{k+1|k} = \tilde{\phi}(\tilde{s}_{k|k}). \tag{10}$$

And we use function $\tilde{\gamma}$ generated by the third deep feedforward neural network (i.e. estimation network) to approximate $\gamma$ in equation 6, i.e.,

$$\hat{x}_{k|k} = \tilde{\gamma}(\tilde{s}_{k|k}). \tag{11}$$

We consider the probability space $(\Omega, \mathscr{F}, \mathbb{P})$ with inner product $\langle x, y \rangle = \mathbb{E}[x^T y]$ and norm $\|x\| := \mathbb{E}^{1/2}[x^T x]$, which is a Hilbert space and denoted as $L^2(\Omega, \mathscr{F}, \mathbb{P})$. We first state the universal approximation theorem of feedforward neural network before we proceed to show our results.

**Theorem 2** (Universal Approximation Theorem (Hornik et al. (1989))). *For any given compact subset $K \subset R^n$, any given continuous function $f$ defined on $K$ and any given accuracy degree $\epsilon > 0$, there exists a function $g$ represented by a single-hidden-layer neural network with non-constant and bounded activation function such that $\max_{x \in K} |f(x) - g(x)| < \epsilon$.*

*Proof.* It is a natural corollary of the Thm. 2.1 in Hornik et al. (1989). $\qquad \square$

Define $e_{k|k} := \|s_{k|k} - \tilde{s}_{k|k}\|$, which represents the *cumulative error* caused by the approximation error of $\tilde{\phi}$ and $\tilde{\varphi}$. Similarly, we define $e_{k|k-1} := \|s_{k|k-1} - \tilde{s}_{k|k-1}\|$. In the following theorem, we shall give the condition which ensures the cumulative error will not blow up as time $k$ approaches $\infty$.

Before we proceed to show our main result, we first establish a key lemma, Lem. 2. We make two assumptions on the system we'll consider..

**Assumption 1.** *We assume that the linear dynamic system of the state in equation 7 is stable in mean square sense (Samuels (1959)), i.e.,*

$$\varlimsup_{k \to \infty} \|x_k\| \le M, \tag{12}$$

*where $M$ is a finite constant.*

**Assumption 2.** *The dynamical system equation 7 is uniformly completely observable and uniformly completely controllable.*

The definitions of uniformly completely observable and uniformly completely controllable can be found in section 7.5 of Jazwinski (1970). We then state a lemma on the boundedness of conditional covariance.

**Lemma 1** (Lemma 7.1 in Jazwinski (1970)). *If Assumption 2 is satisfied and $P_{0|0} \succcurlyeq 0^3$, then $P_{k|k}$ is uniformly bounded from above for all $k \geq N$,*

$$P_{k|k} \preccurlyeq \left( \frac{1 + \alpha\beta}{\alpha} \right) I, \ k \geq N, \tag{13}$$

*where $N$ is a positive integer, $I$ is the $n \times n$ identity matrix and $\alpha$, $\beta$ are positive constants.*

Based on Lemma 1, Assumption 1 and Assumption 2, we give the key Lemma 2.

**Lemma 2.** *In the discrete linear system equation 7, suppose the Assumption 1 and Assumption 2 are satisfied, then for any given $\epsilon > 0$ there exists a compact subset $K \subset \mathrm{R}^{\dim(s_{k|k})}$ such that the statistics computed by KF $s_{k|k-1}, s_{k|k}$ and the statistics $\tilde{s}_{k|k-1}, \tilde{s}_{k|k}$ computed by RNN based filter with non-constant and bounded activation function satisfy $\left\| s_{k|k-1} \mathbb{1}_{s_{k|k-1} \notin K} \right\| < \epsilon$, $\left\| s_{k|k} \mathbb{1}_{s_{k|k} \notin K} \right\| < \epsilon$, $\left\| \tilde{s}_{k|k-1} \mathbb{1}_{\tilde{s}_{k|k-1} \notin K} \right\| < \epsilon$ and $\left\| \tilde{s}_{k|k} \mathbb{1}_{\tilde{s}_{k|k} \notin K} \right\| < \epsilon$, where $\mathbb{1}_A$ is an indicator function.*

*Proof.* Proof can be found in our appendix A.4. ∎

We then derive our main result.

**Theorem 3.** *Assume $s_{k|k}, k \geq 0$ are the theoretical statistics evolving according to equation 9 and $\tilde{s}_{k|k}, k \geq 0$ are the real statistics computed by our RNN-based filter evolving according to equation 10. Suppose the Assumption 1 and Assumption 2 are satisfied. Furthermore, we need to assume functions $\varphi$, $\phi$, $\gamma$ are Lipschitz, i.e., for any $s_1, s_2$,*

$$\begin{aligned}
\|\varphi(s_1, y) - \varphi(s_2, y)\| &\leq C_\varphi \|s_1 - s_2\|, \\
\|\phi(s_1) - \phi(s_2)\| &\leq C_\phi \|s_1 - s_2\|, \\
\|\gamma(s_1) - \gamma(s_2)\| &\leq C_\gamma \|s_1 - s_2\|,
\end{aligned} \tag{14}$$

*where $C_\varphi$ and $C_\phi$ are Lipschitz constants. If $C_\varphi$ and $C_\phi$ satisfy $|C_\varphi C_\phi| < 1$, then for any $\epsilon > 0$, there exists an RNN based filter (with non-constant and bounded activation function) such that*

$$\limsup_{k \to \infty} e_{k|k} = \limsup_{k \to \infty} \left\| s_{k|k} - \tilde{s}_{k|k} \right\| < \epsilon. \tag{15}$$

*Furthermore, we have*

$$\limsup_{k \to \infty} \left\| \hat{x}_{k|k} - \mathbb{E}[x_k | Y_k] \right\| < \epsilon. \tag{16}$$

*Proof.* For any $\delta > 0$, we have the following. By Lem. 3, there exists a compact ball $K = \mathrm{B}(0, r) \subset \mathrm{R}^{\dim(s_{k|k})}$, such that $\left\| s_{k|k-1} \mathbb{1}_{s_{k|k-1} \notin K} \right\| < \delta$, $\left\| s_{k|k} \mathbb{1}_{s_{k|k} \notin K} \right\| < \delta$, $\left\| \tilde{s}_{k|k-1} \mathbb{1}_{\tilde{s}_{k|k-1} \notin K} \right\| < \delta$ and $\left\| \tilde{s}_{k|k} \mathbb{1}_{\tilde{s}_{k|k} \notin K} \right\| < \delta$. By Theorem 2, given any small $\delta_\varphi, \delta_\phi \in \mathbb{R}^+$ and $\delta_\gamma \in \mathbb{R}^+$, there exist two functions $\tilde{\varphi}, \tilde{\phi}$ which are represented by the DNN, such that

$$\|\varphi - \tilde{\varphi}\|_\infty^K \leq \delta_\varphi, \ \|\phi - \tilde{\phi}\|_\infty^K \leq \delta_\phi, \ \|\gamma - \tilde{\gamma}\|_\infty^K \leq \delta_\gamma. \tag{17}$$

where $\|h\|_\infty^K := \max_{x \in K} |h(x)|$. And without loss of generality, we set $\phi(0) = \tilde{\phi}(0)$, $\varphi(0) = \tilde{\varphi}(0)$ and $\gamma(0) = \tilde{\gamma}(0)$. In the prediction step, based on the evolution equations equation 9 and equation 10, we have

$$\begin{aligned}
e_{k|k-1} = \|(s_{k|k-1} - \tilde{s}_{k|k-1})\| &= \|\phi(s_{k-1|k-1}) - \tilde{\phi}(\tilde{s}_{k-1|k-1})\| \\
&\leq \|\phi(s_{k-1|k-1}) - \phi(\tilde{s}_{k-1|k-1})\| + \|\phi(\tilde{s}_{k-1|k-1}) - \tilde{\phi}(\tilde{s}_{k-1|k-1})\| \\
&\leq \|\phi(s_{k-1|k-1}) - \phi(\tilde{s}_{k-1|k-1})\| + \|(\phi(\tilde{s}_{k-1|k-1}) - \tilde{\phi}(\tilde{s}_{k-1|k-1})) \mathbb{1}_{\tilde{s}_{k-1|k-1} \in K}\| \\
&\quad + \|(\phi(\tilde{s}_{k-1|k-1}) - \tilde{\phi}(\tilde{s}_{k-1|k-1})) \mathbb{1}_{\tilde{s}_{k-1|k-1} \notin K}\| \\
&\leq C_\phi e_{k-1|k-1} + \delta_\phi + (C_\phi + C_{\tilde{\phi}}) \delta,
\end{aligned} \tag{18}$$

---

[3]Here, $X \succcurlyeq Y$ ($X \preccurlyeq Y$ resp.) if and only if $X - Y$ ($Y - X$ resp.) is positive semi-definite, where $X$ and $Y$ are symmetric matrices.

where the last inequality follows from equation 14 and equation 17 and $C_{\tilde{\phi}}$ is the Lipschitz constant of $\tilde{\phi}$. We let $\delta'_\phi := \delta_\phi + (C_\phi + C_{\tilde{\phi}})\delta$.

Similarly, in the updating step, we have $e_{k|k} \leq C_\varphi e_{k|k-1} + \delta'_\varphi$ ,where $\delta'_\varphi := \delta_\varphi + (C_\varphi + C_{\tilde{\varphi}})\delta$. Combining this and equation 18, we obtain

$$e_{k|k} \leq C_\varphi e_{k|k-1} + \delta'_\varphi \leq (C_\varphi C_\phi) e_{k-1|k-1} + \left(C_\varphi \delta'_\phi + \delta'_\varphi\right). \tag{19}$$

Using equation 19 repeatedly, it follows that [4]

$$e_{k|k} \leq (C_\varphi C_\phi)^k e_{0|0} + \left(C_\varphi \delta'_\phi + \delta'_\varphi\right) \frac{(C_\varphi C_\phi)^k - 1}{C_\varphi C_\phi - 1}. \tag{20}$$

Thus $\limsup_{k \to +\infty} e_{k|k} \leq \left(C_\varphi \left(\delta_\phi + (C_\phi + C_{\tilde{\phi}})\delta\right) + \delta_\varphi + (C_\varphi + C_{\tilde{\varphi}})\delta\right) \frac{1}{1 - C_\varphi C_\phi}$ as $k \to \infty$ once the condition $|C_\varphi C_\phi| < 1$ holds. We choose small enough $\delta_\phi$, $\delta_\varphi$ and $\delta$ such that $\left(C_\varphi \left(\delta_\phi + (C_\phi + C_{\tilde{\phi}})\delta\right) + \delta_\varphi + (C_\varphi + C_{\tilde{\varphi}})\delta\right) \frac{1}{1 - C_\varphi C_\phi} < \epsilon$. Then we get 15. Now we prove 16.

$$\left\|\hat{x}_{k|k} - \mathbb{E}[x_k|Y_k]\right\| = \left\|\tilde{\gamma}(\tilde{s}_{k|k}) - \gamma(s_{k|k})\right\| \leq \left\|\gamma(s_{k|k}) - \gamma(\tilde{s}_{k|k})\right\| + \left\|\gamma(\tilde{s}_{k|k}) - \tilde{\gamma}(\tilde{s}_{k|k})\right\|$$

$$\leq \left\|\gamma(s_{k|k}) - \gamma(\tilde{s}_{k|k})\right\| + \left\|\left(\gamma(\tilde{s}_{k|k}) - \tilde{\gamma}(\tilde{s}_{k|k})\right) \mathbb{1}_{\tilde{s}_{k-1|k-1} \in K}\right\| + \left\|\left(\gamma(\tilde{s}_{k|k}) - \tilde{\gamma}(\tilde{s}_{k|k})\right) \mathbb{1}_{\tilde{s}_{k-1|k-1} \notin K}\right\|$$

$$\overset{\star_1}{\leq} C_\gamma e_{k|k} + \delta_\gamma + (C_\gamma + C_{\tilde{\gamma}})\delta \overset{\star_2}{\leq} C_\gamma (C_\varphi C_\phi)^k e_{0|0} + C_\gamma \left(C_\varphi \delta'_\phi + \delta'_\varphi\right) \frac{(C_\varphi C_\phi)^k - 1}{C_\varphi C_\phi - 1} + \delta_\gamma + (C_\gamma + C_{\tilde{\gamma}})\delta \tag{21}$$

where the inequality $\star_1$ follows equation 14 and equation 17, the inequality $\star_2$ follows equation 59, and $C_{\tilde{\gamma}}$ is the Lipschitz constant of $\tilde{\gamma}$. Thus $\limsup_{k \to +\infty} \left\|\hat{x}_{k|k} - \mathbb{E}[x_k|Y_k]\right\| \leq C_\gamma (C_\varphi \delta'_\phi + \delta'_\varphi)(-C_\varphi C_\phi + 1)^{-1} + \delta_\gamma + (C_\gamma + C_{\tilde{\gamma}})\delta$. Again we can choose small enough $\delta_\phi$, $\delta_\varphi$, $\delta_\gamma$ and $\delta$ such that $C_\gamma (C_\varphi \delta'_\phi + \delta'_\varphi)(-C_\varphi C_\phi + 1)^{-1} + \delta_\gamma + (C_\gamma + C_{\tilde{\gamma}})\delta < \epsilon$. Then we obtain the desired 16. $\square$

An example satisfying all the assumptions of Thm. 3 can be found in our Appendix A.5, and a general class of systems satisfying assumption $|C_\varphi C_\phi| < 1$ can be found in Appendix A.6. We also remark that in our proof, we implicitly require that the Lipschitz constants of $\tilde{\phi}, \tilde{\varphi}, \tilde{\gamma}$ are uniformly upper bounded. (See more details in our Appendix A.8) Thm. 3 highlights that the optimal filter in linear system with Gaussian noise, Kalman Filter, can be synthesized by RNN. And RNN based filter's asymptotic error can be as small as wanted under some Lipschitz conditions. That is to say, RNN is an *approximator* of Kalman filter.

## 4.2 RNN BASED FILTER IS A UNIVERSAL APPROXIMATOR OF OPTIMAL FINITE DIMENSIONAL FILTER

Thm. 3 shows that Kalman Filter (KF) can be synthesized by RNN. In this section, we'll try to answer the question "How general is the RNN based filter?" and extend the result into a more general case. We'll show that any optimal finite dimensional filter can be *universally approximated* by RNN under some mild conditions. For a general system with noisy observation as shown in equation 1, once the conditional distribution $p(x_k|Y_k)$ is obtained, the filtering problem is solved. However, we usually need to solve an infinite number of ordinary differential equations (ODE) in order to solve $p(x_k|Y_k)$. If the distribution $p(x_k|Y_k)$ admits a finite dimensional sufficient statistics, then we only need to solve a finite number of ODE (Chen (2003)) and we call such filter *finite dimensional filter*. Finite dimensional filter has been an active research area after the seminal work (Beneš (1981; 1985)) of Beneš. It's is a large class of filters. Some nontrivial finite dimensional nonlinear filter examples can be found in Daum (1986); Ferrante & Runggaldier (1990); GÜnther (1981); Levine & Marino (1986).

Similarly, we use vector $S_{k|k}$ to denote the finite dimensional sufficient statistics of the posterior distribution $p(x_k|Y_k)$ and $S_{k|k-1}$ to denote the finite dimensional sufficient statistics of the prior

---

[4]See more details in our Appendix A.7.

distribution $p(x_k|Y_{k-1})$. The evolution functions of the statistics are denoted as $\Phi$ and $\Psi$, and the map from $S_{k|k}$ to conditional mean $\mathbb{E}(x_k|Y_k)$ is denoted as $\Gamma$, i.e.,

$$S_{k|k-1} = \Phi(S_{k-1|k-1}), \ S_{k|k} = \Psi(S_{k|k-1}, y_k), \ \mathbb{E}(x_k|Y_k) = \Gamma(S_{k|k}). \tag{22}$$

Similarly, in our proposed neural networks, we use DNN generated function $\tilde{\Phi}$ (prediction network) to approximate $\Phi$, use another DNN generated $\tilde{\Psi}$ (update network) to approximate $\Psi$, and use the third DNN generated function $\tilde{\Gamma}$ to approximate $\Gamma$. And the numerical statistics computed by RNN are denoted as $\tilde{S}_{k|k}$ and $\tilde{S}_{k|k-1}$, i.e.,

$$\tilde{S}_{k|k-1} = \tilde{\Phi}(\tilde{S}_{k-1|k-1}), \ \tilde{S}_{k|k} = \tilde{\Psi}(\tilde{S}_{k|k-1}, y_k), \ \hat{x}_{k|k} = \tilde{\Gamma}(\tilde{S}_{k|k}). \tag{23}$$

We also need the following assumption.

**Assumption 3.** *We assume that for any given $\epsilon > 0$ there exists a compact subset $K \subset \mathrm{R}^{\dim(S_{k|k})}$ such that the statistics computed by the optiaml finite dimensional filter $S_{k|k-1}$ and $S_{k|k}$ satisfy $\left\| S_{k|k-1} \mathbb{1}_{S_{k|k-1} \notin K} \right\| < \epsilon$, and $\left\| S_{k|k} \mathbb{1}_{S_{k|k} \notin K} \right\| < \epsilon$.*

We can see equation 7 is the special case of system equation 1, and it satisfies the Assumption 3 according to Lemma 2. We further have Lem. 3 and Thm. 4.

**Lemma 3.** *In the discrete system equation 1, for any given $\epsilon > 0$ there exists a compact subset $K \subset \mathrm{R}^{\dim(S_{k|k})}$ such that the statistics $\tilde{S}_{k|k-1}, \tilde{S}_{k|k}$ computed by RNN based filter with non-constant and bounded activation function satisfy $\left\| \tilde{S}_{k|k-1} \mathbb{1}_{\tilde{S}_{k|k-1} \notin K} \right\| < \epsilon$ and $\left\| \tilde{S}_{k|k} \mathbb{1}_{\tilde{S}_{k|k} \notin K} \right\| < \epsilon$, where $\mathbb{1}_A$ is indicator function.*

*Proof.* The proof is similar to the step 2 of the proof of Lem. 2. $\qquad\square$

**Theorem 4.** *Consider a discrete filtering system equation 1 with optimal finite dimensional filter and suppose $S_{k|k}, k \geq 0$ are the theoretical statistics evolving according to equation 22 and $\tilde{S}_{k|k}, k \geq 0$ are the statistics generated by our RNN based filter and evolving according to equation 23. Suppose the Assumption 3 is satisfied. Furthermore, we need to assume that functions $\Phi$ and $\Psi$ are Lipschitz, i.e., for any $S_1, S_2$,*

$$\begin{aligned} \|\Psi(S_1, y) - \Psi(S_2, y)\| &\leq C_\Psi \|S_1 - S_2\|, \\ \|\Phi(S_1) - \Phi(S_2)\| &\leq C_\Phi \|S_1 - S_2\|, \\ \|\Gamma(S_1) - \Gamma(S_2)\| &\leq C_\Gamma \|S_1 - S_2\| \end{aligned} \tag{24}$$

*where $C_\Psi$ and $C_\Phi$ are Lipschitz constants. If $C_\Psi$ and $C_\Phi$ satisfy $|C_\Psi C_\Phi| < 1$, then for any $\epsilon > 0$, there exists a RNN based filter with non-constant and bounded activation function such that*

$$\limsup_{k \to \infty} \left\| S_{k|k} - \tilde{S}_{k|k} \right\| < \epsilon. \tag{25}$$

*Furthermore, we have*

$$\limsup_{k \to \infty} \left\| \hat{x}_{k|k} - \mathbb{E}[x_k|Y_k] \right\| < \epsilon. \tag{26}$$

*Proof.* The procedure is similar to the proof of Theorem 3. $\qquad\square$

Thm. 4 highlights that RNN based filter can not only approximate Kalman filter, but any optimal finite dimensional filter under some Lipschitz conditions. Therefore, RNN's expressive power is characterized as the *universal filtering property*.

## 5 CONCLUSION

In our paper, we try to characterize the expressive power of RNN from the filtering perspective. We unify the recurrent neural network into Bayesian filtering framework and show that recurrent neural network is a *universal approximator of optimal finite dimensional filters* under some Lipschitz conditions. As an interesting special case, the widely used Kalman filter can be *synthesized by RNN*. Understanding the expressive power of RNN based filter in more general nonlinear filtering cases (with no finite dimensional sufficient statistics) can be a very interesting future direction.

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

# A  APPENDIX

## A.1  PROOF OF THEOREM 1

*Proof.* It is known that

$$
\begin{aligned}
\mathbb{E}[L_{MSE}] &= \mathbb{E}[(x_k - \hat{x}_k)^T(x_k - \hat{x}_k)] \\
&= \mathbb{E}\left[\mathbb{E}\left[(x_k - \hat{x}_k)^T(x_k - \hat{x}_k)\big| Y_k\right]\right],
\end{aligned}
\tag{27}
$$

and let

$$
\mu_k = \mathbb{E}[x_k|Y_k].
$$

Since

$$
\begin{aligned}
&\mathbb{E}\left[(x_k - \mu_k)^T(\mu_k - \hat{x}_k)\big| Y_k\right] \\
&= (x_k - \mu_k)^T(\mu_k - \mu_k) = 0,
\end{aligned}
\tag{28}
$$

and

$$
\mathbb{E}\left[(x_k - \mu_k)^T(x_k - \mu_k)\right]
\tag{29}
$$

is independent of $\hat{x}_k$, equation 27 becomes

$$
\begin{aligned}
\mathbb{E}[L_{MSE}] &= \mathbb{E}\left[\mathbb{E}\left[(x_k - \hat{x}_k)^T(x_k - \hat{x}_k)\big| Y_k\right]\right] \\
&= \mathbb{E}\left[\mathbb{E}\left[(x_k - \mu_k + \mu_k - \hat{x}_k)^T(x_k - \mu_k + \mu_k - \hat{x}_k)\big| Y_k\right]\right] \\
&= \mathbb{E}\left[\mathbb{E}\left[(x_k - \mu_k)^T(x_k - \mu_k) + (\mu_k - \hat{x}_k)^T(\mu_k - \hat{x}_k)\right.\right. \\
&\quad \left.\left. + 2(x_k - \mu_k)^T(\mu_k - \hat{x}_k)\big| Y_k\right]\right] \\
&= \mathbb{E}\left[(x_k - \mu_k)^T(x_k - \mu_k)\right] + \mathbb{E}\left[(\mu_k - \hat{x}_k)^T(\mu_k - \hat{x}_k)\right] \\
&= \text{const} + \mathbb{E}\left[(\mu_k - \hat{x}_k)^T(\mu_k - \hat{x}_k)\right]
\end{aligned}
\tag{30}
$$

Therefore $\mathbb{E}[L_{MSE}]$ is clearly minimized by setting $\hat{x}_k = \mu_k = \mathbb{E}[x_k|Y_k]$. $\qquad\square$

## A.2 THE EVOLUTION OF DISTRIBUTION IN UPDATING STEP OF BAYESIAN FILTER

Let $p(x_k|Y_k)$ denote the conditional distribution of state $x_k$ conditioned on the observation history $Y_k$, then using Bayes' rule, we have:

$$
\begin{aligned}
p(x_k|Y_k) &\overset{\text{Bayes' rule}}{=} \frac{p(Y_k|x_k)p(x_k)}{p(Y_k)} = \frac{p(y_k, Y_{k-1}|x_k)p(x_k)}{p(y_k, Y_{k-1})} \\
&= \frac{p(y_k|Y_{k-1}, x_k)p(Y_{k-1}|x_k)p(x_k)}{p(y_k|Y_{k-1})p(Y_{k-1})} \\
&\overset{\text{Bayes' rule}}{=} \frac{p(y_k|Y_{k-1}, x_k)p(x_k|Y_{k-1})p(Y_{k-1})p(x_k)}{p(y_k|Y_{k-1})p(Y_{k-1})p(x_k)} \\
&= \frac{p(y_k|x_k)p(x_k|Y_{k-1})}{p(y_k|Y_{k-1})} \\
&= \frac{p(y_k|x_k)p(x_k|Y_{k-1})}{\int p(y_k|x_k)p(x_k|Y_{k-1})\,\mathrm{d}x_k}.
\end{aligned}
\tag{31}
$$

It can be seen that the posterior distribution $p(x_k|Y_k)$ is determined by the prior distribution $p(x_k|Y_{k-1})$ and likelihood function $p(y_k|x_k)$, which depends on the observation model and the known distribution of $v_k$.

## A.3 THE EVOLUTION FUNCTION OF STATISTICS FOR KALMAN FILTER

The widely used Kalman Filter assumes that the posterior distribution at every instant is Gaussian, and then it can be parameterized by the mean and covariance. We consider the following system:

$$
\begin{cases}
x_k = Fx_{k-1} + Gw_{k-1} \\
y_k = Hx_k + v_k,
\end{cases}
\tag{32}
$$

where $F, G, H$ are constant matrices with proper dimensions, the initial state $x_0$ is Gaussian, and $\{w_k, k = 0, 1, \cdots\}$ and $\{v_k, k = 1, \cdots\}$ are two independent white Gaussian sequences that are also independent of the initial state $x_0$ jointly.

Then KF can be viewed as the following recursive relationship:

$$
\begin{aligned}
p(x_{k-1}|Y_{k-1}) &= \mathcal{N}(x_{k-1}; m_{k-1|k-1}, P_{k-1|k-1}) \\
p(x_k|Y_{k-1}) &= \mathcal{N}(x_k; m_{k|k-1}, P_{k|k-1}) \\
p(x_k|Y_k) &= \mathcal{N}(x_k; m_{k|k}, P_{k|k}),
\end{aligned}
\tag{33}
$$

where in the prediction step,

$$
\begin{cases}
m_{k|k-1} = Fm_{k-1|k-1} \\
P_{k|k-1} = GQG^T + FP_{k-1|k-1}F^T,
\end{cases}
\tag{34}
$$

in the updating step,

$$
\begin{cases}
m_{k|k} = m_{k|k-1} + P_{k|k-1}H^T(HP_{k|k-1}H^T + R)^{-1} \\
\qquad\qquad \cdot (y_k - Hm_{k|k-1}) \\
P_{k|k} = P_{k|k-1} - P_{k|k-1}H^T(HP_{k|k-1}H^T + R)^{-1} \\
\qquad\qquad \cdot HP_{k|k-1},
\end{cases}
\tag{35}
$$

and $\mathcal{N}(x; m, P)$ is a Gaussian distribution with mean $m$, and covariance $P$. The evolution functions of mean and covariance in the prediction step and updating step can be expressed in the following compact form:

$$
\begin{cases}
m_{k|k-1} = \phi_1(m_{k-1|k-1}, P_{k-1|k-1}) \\
P_{k|k-1} = \phi_2(m_{k-1|k-1}, P_{k-1|k-1}),
\end{cases}
\tag{36}
$$

$$\begin{cases} m_{k|k} = \varphi_1(m_{k|k-1}, P_{k|k-1}, y_k) \\ P_{k|k} = \varphi_2(m_{k|k-1}, P_{k|k-1}, y_k), \end{cases} \tag{37}$$

where $\phi_1, \phi_2, \varphi_1$ and $\varphi_2$ are defined as

$$\begin{cases} \phi_1(m, P) := Fm \\ \phi_2(m, P) := GQG^T + FPF^T, \end{cases} \tag{38}$$

and

$$\begin{cases} \varphi_1(m, P, y) := m + PH^T(HPH^T + R)^{-1}(y - Hm) \\ \varphi_2(m, P, y) := P - PH^T(HPH^T + R)^{-1}HP, \end{cases} \tag{39}$$

Here,

$$m_{k|k-1} = \mathbb{E}[x_k|Y_{k-1}], \ m_{k|k} = \mathbb{E}[x_k|Y_k] \tag{40}$$

are the conditional means of state conditioned on observation history.

$$\begin{aligned} P_{k|k-1} &= \mathbb{E}\left[(x_k - m_{k|k-1})(x_k - m_{k|k-1})^T|Y_{k-1}\right], \\ P_{k|k} &= \mathbb{E}\left[(x_k - m_{k|k})(x_k - m_{k|k})^T|Y_k\right] \end{aligned} \tag{41}$$

are the corresponding conditional covariance. Then we know that the posterior distribution $p(x_k|Y_k)$ is parameterized by the conditional mean $m_{k|k}$ and covariance $P_{k|k}$ which are functions of $Y_k$, i.e., $p(x_k|Y_{k-1})$ is parameterized by $\{m_{k|k}, \ P_{k|k}\}$. And the similar conclusion can be obtained for prior distribution $p(x_k|Y_{k-1})$.

Now we vectorize $\{m_{k|k}, \ P_{k|k}\}$ as the statistics

$$s_{k|k} := [m_{k|k}{}^T, \text{vec}^T(P_{k|k})]^T. \tag{42}$$

Vectorizing the second equation in equation 36 and equation 37, we have

$$\begin{aligned} \begin{bmatrix} \varphi_1(m, P, y) \\ \text{vec}(\varphi_2(m, P, y)) \end{bmatrix} &= \begin{bmatrix} \varphi_1(m, \text{vec}^{-1}(\text{vec}(P)), y) \\ \text{vec}(\varphi_2(m, \text{vec}^{-1}(\text{vec}(P)), y)) \end{bmatrix} \\ &\triangleq \varphi(m, \text{vec}(P), y) = \varphi(s, y), \end{aligned} \tag{43}$$

and

$$\begin{bmatrix} \phi_1(m, P) \\ \text{vec}(\phi_2(m, P)) \end{bmatrix} \triangleq \phi(m, \text{vec}(P)) = \phi(s), \tag{44}$$

where we omit subscript characters without causing confusion, and $\text{vec}_{n_1 \times n_2}^{-1}$ is the inverse operator of vec such that $\text{vec}_{n_1 \times n_2}^{-1}(\text{vec}(\circ_{n_1 \times n_2})) = \circ_{n_1 \times n_2}$. Then we obtain the evolution functions of the statistics:

$$s_{k|k} = \varphi(s_{k|k-1}, y_k), \ s_{k+1|k} = \phi(s_{k|k}), \tag{45}$$

where $\varphi$ and $\phi$ are defined in equation 43 and equation 44.

## A.4  PROOF TO LEM. 2

*Proof.* **Step 1:** It can be known from the KF that the evolution function for conditional covariance is as follows (see appendix):

$$\begin{cases} P_{k|k-1} = GQG^T + FP_{k-1|k-1}F^T \\ P_{k|k} = P_{k|k-1} - P_{k|k-1}H^T(HP_{k|k-1}H^T + R)^{-1} \\ \qquad\qquad \cdot HP_{k|k-1}, \end{cases} \tag{46}$$

and the initial value is $P_{0|0}$. It follows that the conditional covariance $P$ evolves in a deterministic manner and without any randomness according to the evolution equation equation 46, therefore $P$ is independent of the observations. Since

$$P_{k|k} = \mathbb{E}\left[\left(x_k - m_{k|k}\right)\left(x_k - m_{k|k}\right)^T\middle|Y_k\right],$$

and $P_{k|k}$ is independent of $Y_k$, we have

$$P_{k|k} = \mathbb{E}\left[\left(x_k - m_{k|k}\right)\left(x_k - m_{k|k}\right)^T\right]. \tag{47}$$

According to Lemma 1, we have

$$P_{k|k} \preccurlyeq \left(\frac{1 + \alpha\beta}{\alpha}\right) I, \; k \geq N, \tag{48}$$

where $N$ is a positive integer and $\alpha$, $\beta$ are positive constants. It can be easily checked that, there exists a positive constant $\alpha_0$, such that

$$P_{k|k} \preccurlyeq \alpha_0 I, \; \forall \, k \geq 0. \tag{49}$$

According to equation 47, we have

$$
\begin{aligned}
&\|x_k - m_{k|k}\|^2 \\
=&\mathbb{E}\left[\left(x_k - m_{k|k}\right)^T\left(x_k - m_{k|k}\right)\right] \\
=&\text{tr}\left(P_{k|k}\right),
\end{aligned} \tag{50}
$$

where $\text{tr}(\star)$ denotes the trace of matrix $\star$. Combining equation 49 and equation 50, we know that

$$\|x_k - m_{k|k}\| \leq \sqrt{n\alpha_0}, \; \forall \, k \geq 0. \tag{51}$$

Then according to equation 12 and equation 51, we have

$$
\begin{aligned}
\|m_{k|k}\| &\leq \|x_k - m_{k|k}\| + \|x_k\| \\
&\leq \sqrt{n\alpha_0} + M \\
&\leq C_0, \; \forall \, k \geq 0,
\end{aligned} \tag{52}
$$

where $C_0$ is a positive constant and can be chosen to be $\sqrt{n\alpha_0} + M$.

Now we have the conclusion that $m_{k|k}$ and $P_{k|k}$ are bounded for all $k \geq 0$. Similarly, we know that $m_{k|k-1}$ and $P_{k|k-1}$ are all bounded following the similar procedure as above. According to equation 8, we know that all sufficient statistics $s_{k|k-1}$, $s_{k|k}$ are bounded.

**Step 2:** We then prove that $\left\|\tilde{s}_{k|k-1}\mathbb{1}_{\tilde{s}_{k|k-1}\notin K}\right\| < \epsilon$ and $\left\|\tilde{s}_{k|k}\mathbb{1}_{\tilde{s}_{k|k}\notin K}\right\| < \epsilon$. Recall that

$$\tilde{s}_{k|k} = \tilde{\varphi}(\tilde{s}_{k|k-1}, y_k), \; \tilde{s}_{k+1|k} = \tilde{\phi}(\tilde{s}_{k|k}), \tag{53}$$

where $\tilde{\varphi}$ and $\tilde{\phi}$ are approximating functions generated by neural networks with non-constant and bounded activation functions. Let $\sigma$ denote the activation functions used by the neural network and assume that $|\sigma(x)| < C_\sigma, \forall x \in \mathbb{R}$. And without loss of generality, we assume $\tilde{\varphi}(\tilde{s}_{k|k-1}, y_k) = V_1\sigma(W_1 p_1(\tilde{s}_{k|k-1}, y_k) + b_1) + c_1$, where $p_1$ is the vector valued function that takes input to the last hidden layer's output, $V_1$ and $W_1$ are matrices with suitable dimension, $b_1$ is a vector with suitable dimension. Then we'll have,

$$
\begin{aligned}
&\left\|\tilde{s}_{k|k}\right\| \\
=&\left\|\tilde{\varphi}(\tilde{s}_{k|k-1}, y_k)\right\| \\
\leq&\left\|V_1\sigma(W_1 p_1(\tilde{s}_{k|k-1}, y_k) + b_1) + c_1\right\| \\
\leq&\left\|V_1\right\|\left\|\sigma(W_1 p_1(\tilde{s}_{k|k-1}, y_k) + b_1)\right\| + \left\|c_1\right\| \\
\leq&\left\|V_1\right\|\sqrt{\dim(b_1)}C_\sigma + \left\|c_1\right\|,
\end{aligned} \tag{54}
$$

where $\dim(x)$ is the dimension of the vector of $x$. We choose the compact set $K$ to cover the compact ball $\{x \in \mathbb{R}^{\dim(\tilde{s}_{k|k})} : \|x\|_E \leq \|V_1\|\sqrt{\dim(b_1)}C_\sigma + \|c_1\|\}$ where $\|\star\|_E$ denotes Euclidean norm of $\star$. Then it can be seen that $\left\|\tilde{s}_{k|k-1}\mathbb{1}_{\tilde{s}_{k|k-1}\notin K}\right\| < \epsilon$. And $\left\|\tilde{s}_{k|k}\mathbb{1}_{\tilde{s}_{k|k}\notin K}\right\| < \epsilon$ can be shown in a similar way. $\qquad\square$

## A.5    AN EXAMPLE THAT SATISFIES ALL THE ASSUMPTIONS OF THM. 3

Here we verify that the system shown in equation 55 with state's dimension 1 satisfies all assumptions in Theorem 3.

$$\begin{cases} x_k = (1-\alpha)x_{k-1} + \sqrt{\alpha}w_{k-1} \\ y_k = \alpha x_k + \sqrt{\alpha}v_k, \end{cases} \tag{55}$$

where $0 < \alpha < 1$ is a small positive parameter.

- As for Assumption 1, from the state equation of equation 55 we have

$$\begin{aligned} x_k =& (1-\alpha)x_{k-1} + \sqrt{\alpha}w_{k-1} \\ =& (1-\alpha)^2 x_{k-2} + (1-\alpha)\sqrt{\alpha}w_{k-2} + \sqrt{\alpha}w_{k-1} \\ & \vdots \\ =& (1-\alpha)^k x_0 + \sum_{i=0}^{k-1}(1-\alpha)^{k-1-i}\sqrt{\alpha}w_i, \end{aligned} \tag{56}$$

then we have

$$\begin{aligned} \mathbb{E}[|x_k|^2] =& (1-\alpha)^{2k}\mathbb{E}[|x_0|^2] + \sum_{i=0}^{k-1}(1-\alpha)^{2k-2-2i}\alpha Q \\ =& (1-\alpha)^{2k}\mathbb{E}[|x_0|^2] + \frac{(1-\alpha)^{2k-2} - (1-\alpha)^2}{(1-\alpha)^2 - 1}\alpha Q. \end{aligned} \tag{57}$$

It can be easily checked that Assumption 1 is satisfied.

- As for Assumption 2, it can be easily checked that system shown in equation 55 satifies Assumption 2 using the definitions of uniformly completely observable and uniformly completely controllable in section 7.5 of Jazwinski (1970).

## A.6    A CLASS OF SYSTEM THAT SATISFIES $|C_\varphi C_\phi| < 1$ IN THEOREM 3

Actually this condition $|C_\varphi C_\phi| < 1$ can be satisfied by a general class of systems. Let us consider the 1-dimensional linear systems with discretization in continuous observation equation, i.e.,

$$x_{t_k} = ax_{t_{k-1}} + gw_{t_{k-1}}, \ y_{t_k} = b\Delta t x_{t_k} + v_{t_k}^\Delta, \tag{58}$$

where $a, g, b$ are constants, $\Delta t = t_k - t_{k-1} << 1$ is the step size of sampling, $\{w_k, k = 1, \cdots\}$ is an 1$-$dimensional white Gaussian process and $w_k \sim \mathcal{N}(0, Q)$, and $\{v_{t_k}^\Delta\}$ is 1$-$dimensional white Gaussian process with distribution $\mathcal{N}(0, R/\Delta t)$. According to Jazwinski (1970), we know that the discrete observation system in equation 58 is the sampled system of continuous observations.

Following equation (34)-(35) in Appendix A.2, we have

$$\phi_1(m, P) = am, \phi_2(m, P) = gQg + a^2 P,$$

and

$$\varphi_1(m, P) = m + K(y - b\Delta tm), \varphi_2(m, P) = P - Kb\Delta tP,$$

where $K = Pb(\Delta t)^2((b\Delta t)^2 P\Delta t + R)^{-1}$. It can be easily checked that

$$\nabla\phi = \begin{bmatrix} a & 0 \\ 0 & a^2 \end{bmatrix}, \nabla\varphi = \begin{bmatrix} 1 - Kb\Delta t & (y - \Delta tm)\frac{\partial K}{\partial P} \\ 0 & 1 - Kb\Delta t - b\Delta tP\frac{\partial K}{\partial P} \end{bmatrix},$$

where $\frac{\partial K}{\partial P} = (b\Delta t)^2 R((b\Delta t)^2 P\Delta t + R)^{-2}$. Then we have

$$C_\phi = \max\{a, a^2\}, C_\varphi = E\|\nabla\varphi\|_2 \le E\|\nabla\varphi\|_F = 1 + O(\Delta t),$$

where $\|\cdot\|_F$ is the Frobenius norm.

If $a < 1$, which is a very natural stable condition, then

$$\lim_{\Delta t \to 0}|C_\varphi C_\phi| \le \lim_{\Delta t \to 0}|(a + O(\Delta t))| = a < 1.$$

Apparently, we have $|C_\varphi C_\phi| < 1$ with small step size $\Delta t$.

### A.7 CALCULATION DETAILS OF INEQUALITY 20

$$
\begin{aligned}
e_{k|k} &\leq (C_\varphi C_\phi)\, e_{k-1|k-1} + \left(C_\varphi \delta_\phi^{'} + \delta_\varphi^{'}\right) \\
&\leq (C_\varphi C_\phi)\left(C_\varphi C_\phi e_{k-2|k-2} + \left(C_\varphi \delta_\phi^{'} + \delta_\varphi^{'}\right)\right) + \left(C_\varphi \delta_\phi^{'} + \delta_\varphi^{'}\right) \\
&= (C_\varphi C_\phi)^2\, e_{k-2|k-2} + [C_\varphi C_\phi + 1]\left(C_\varphi \delta_\phi^{'} + \delta_\varphi^{'}\right) \\
&\;\;\vdots \\
&\leq (C_\varphi C_\phi)^k\, e_{0|0} + \left(C_\varphi \delta_\phi^{'} + \delta_\varphi^{'}\right) \sum_{i=0}^{k} (C_\varphi C_\phi)^k \\
&= (C_\varphi C_\phi)^k\, e_{0|0} + \left(C_\varphi \delta_\phi^{'} + \delta_\varphi^{'}\right) \frac{(C_\varphi C_\phi)^k - 1}{C_\varphi C_\phi - 1}.
\end{aligned}
\tag{59}
$$

### A.8 APPROXIMATE $\phi$ AND $\varphi$ USING LIPSCHITZ NEURAL NETWORK

In the proof of Thm. 3, we implicitly require the Lipschitz constants of $\tilde{\phi}$ and $\tilde{\varphi}$ are uniformly bounded by another constant $C$. Note that this assumption won't affect the approximation capability of Deep Neural Network in approximating Lipschitz continuous function $\phi$ and $\varphi$ (Thm. 3 in Anil et al. (2018)). With this requirement, the internal state variables of our proposed RNN based filter is always inside a compact ball independent of the choice of neural network. And we will have $\left(C_\varphi\left(\delta_\phi + (C_\phi + C_{\tilde{\phi}})\delta\right) + \delta_\varphi + (C_\varphi + C_{\tilde{\varphi}})\delta\right)\dfrac{1}{1 - C_\varphi C_\phi} <$ $(C_\varphi\left(\delta_\phi + (C_\phi + C)\delta\right) + \delta_\varphi + (C_\varphi + C)\delta)\dfrac{1}{1 - C_\varphi C_\phi}$, thus we can choose small enough $\delta$, $\delta_\phi$ and $\delta_\varphi$ to make $\left(C_\varphi\left(\delta_\phi + (C_\phi + C_{\tilde{\phi}})\delta\right) + \delta_\varphi + (C_\varphi + C_{\tilde{\varphi}})\delta\right)\dfrac{1}{1 - C_\varphi C_\phi} <$ $(C_\varphi\left(\delta_\phi + (C_\phi + C)\delta\right) + \delta_\varphi + (C_\varphi + C)\delta)\dfrac{1}{1 - C_\varphi C_\phi} < \epsilon.$

### A.9 EXPERIMENTS

#### A.9.1 EXPERIMENT SET UP

We use PyTorch as the deep learning framework to implement our algorithm and run our simulation on CPU clusters with 5 E5-2630/v3/2.40GHz CPUs, each is equipped with 30GB memory. We use the Mean Squared Error(MSE) as the metric to measure the performance of the filter. For subsequent discussion, we make the following definition.

**Definition 3** (PMSE). *For a set of sampled paths $\{\omega_1, \omega_2, \cdots, \omega_N\}$, we define the **Path Mean Squared Error** at time slot $k$ $\mathrm{PMSE}_k$ to be $\frac{1}{N}\sum_{i=1}^{N} \|\hat{x}_k(\omega_i) - x_k(\omega_i)\|_2^2$.*

Further we define the Temporal Path Mean Squared Error TPMSE.

**Definition 4** (TPMSE). *For a set of sampled paths $\{\omega_1, \omega_2, \cdots, \omega_N\}$ and a horizon of simulation $1 : M$, we define the Temporal Path Mean Squared Error (TPMSE) to be $\frac{1}{M}\sum_{k=1}^{M} \mathrm{PMSE}_k$.*

#### A.9.2 TRAINING

Alg.1 shows how we train the RNN based filter.

#### A.9.3 KALMAN FILTER CAN BE SYNTHESIZED BY RNN BASED FILTER EFFICIENTLY

We consider a stable linear filtering system as in equation 60.

$$
\begin{cases}
x_k = (\alpha A_n + I_n)x_{k-1} + \sqrt{\alpha}\, w_{k-1} \\
y_k = \alpha x_k + \sqrt{\alpha}\, v_k,
\end{cases}
\tag{60}
$$

---

**Algorithm 1** Bayesian Filter Net (BFN) training algorithm

---

1: Setting the sampling path number $N$, the number of simulated steps M, the optimization horizon length $l$ and the learning rate $\lambda$
2: Sampling $x_0(\omega_i)$ i.i.d from the initial distribution $\sigma_0(x)$ and set $y_0(\omega_i) = 0$
3: **for** $0 \leq m \leq \frac{M}{l} - 1$ **do**
4:     Initialize $\text{loss}(\theta)$ to be 0
5:     **for** $ml \leq k < (m+1)l$ **do**
6:         **for** $1 \leq i \leq N$ **do**
7:             $x_{k+1}(\omega_i) \leftarrow f(x_k(\omega_i)) + g(x_k(\omega_i))w_{k-1}(\omega_i)$
8:             $y_{k+1}(\omega_i) \leftarrow h(x_{k+1}(\omega_i)) + v_k(\omega_i)$
9:         **end for**
10:        **for** $1 \leq i \leq N$ **do**
11:           Feed the sampled $y_{k+1}(\omega_i)$ to the Bayesian-Filter-Net(BFN) and get the corresponding output $\hat{x}_{k+1}(\omega_i)$
12:           Add the estimation squared error to the loss function $\text{loss}(\theta) \leftarrow \text{loss}(\theta) + \|\hat{x}_{k+1}(\omega_i) - x_{k+1}(\omega_i)\|_2^2$
13:        **end for**
14:     **end for**
15:     Doing back propagation and update the network parameters $\theta \leftarrow \theta - \lambda\nabla_\theta\text{loss}(\theta)$
16: **end for**

---

where $\alpha < 1$ is a small positive parameter controlling the changing rate of the system state, $I_n$ is a $n \times n$ identity matrix and $A_n$ is a matrix with elements $a_{ij}$ satisfying $a_{ij} = 1$, if $i = j + 1$, $-\binom{n}{i-1}$, if $j = n$ and $0$, otherwise.

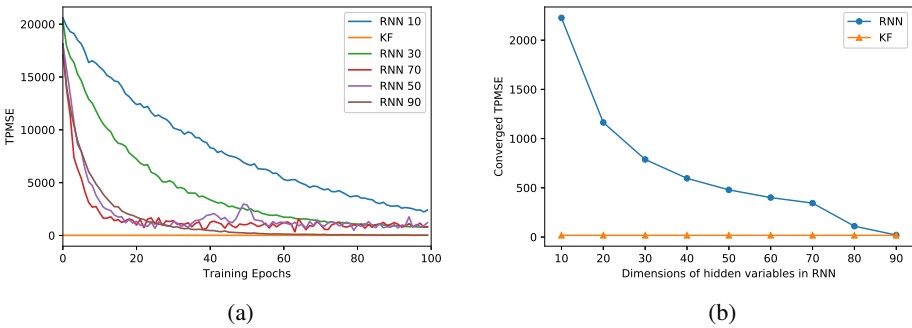

|  | (a) |  | (b) |

Figure 3: (a) The decrease of TPMSE of RNN based filter with different hidden variable dimensions in training. (b) Convergent TPMSE of RNN based filter with different dimensions of hidden variables.

From Fig. 3a, we can see that as the number of training epochs increase, RNN-based filter's MSE decrease and finally converge. And from Fig. 3b as the dimension of the hidden variable increases, the converged MSE decreases. The results in Fig. 3 highlight that just with the number of 90 hidden variables, RNN based filter can achieve the performance of Kalman filter. Considering that there are 65 independent statistics in this case, this is a very efficient approximation.

### A.9.4 RNN BASED FILTER CAN STILL WORK FOR INFINITE DIMENSIONAL FILTER

For finite dimensional filter, our analysis has shown that RNN based filter is a *universal approximator* of optimal filter. In this subsection, we show that in infinite dimensional filter case (where finite dimensional statistics are not admitted to fully determine the distribution), our RNN based method achieves even better performance to state-of-the-art spectral method Luo & Yau (2013). We consider a special and well-known infinite dimensional case.

$$\begin{cases} x_k &= x_{k-1} + \sqrt{\alpha}w_{k-1} \\ y_k &= \alpha x_k^3 + \sqrt{\alpha}v_k \end{cases} \tag{61}$$

where $\alpha$ is a positive parameter controlling the change rate of the system. We set the $\alpha$ to be $0.01$, the total sampling steps to be 5000, the number of sampled paths to be 1000, the number of training epochs to be 100. We only use two fully connected layers for the prediction network, update network and estimation network. Fig. 4a shows the convergence of PMSE of our Deep-Filter-Net with respect to the time. And fig. 4b shows that our proposed Deep-Filter-Net output $\hat{x}_k$ tracks $x_k$ well. We also

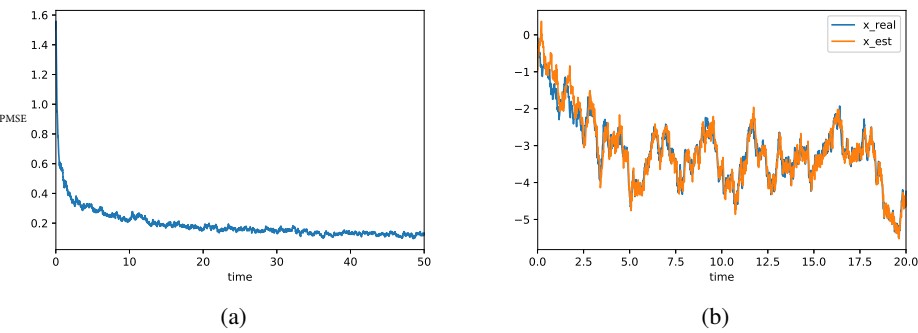

(a)           (b)

Figure 4: (a) The evolution of $\mathrm{PMSE}_k$. (b) $\hat{x}_k$ tracks $x_k$ well.

compare our algorithm's performance with existing state-of-the-art spectral method. Our proposed DFN's TPMSE is 0.18, which is much smaller than spectral method's TPMSE 0.40. One potential reason for its RNN's applicability in infinite dimensional filter case is that RNN implicitly learns the most relevant finite dimensional statistics.

