# OpenReview forum: "Recurrent Neural Networks are Universal Filters"
_ICLR.cc/2020/Conference — Reject_

### Official Review · AnonReviewer3 · 2019-10-23
**Official Blind Review #3**

**Rating:** 3

**Review:**

This paper shows that RNN (of infinite horizon) can be universal approximators  for any stochastic dynamics system. It does that by using the Bayesian framework of filtering, which shows that if sufficient statistics given the observations so far can be computed at each stage of filtering, then the expected hidden states at each stage can be estimated as well. The paper then follows this framework by constructing deep neural network mapping from “observations so-far” to “sufficient statistics” and by universal approximation theorem this is possible. Then as long as after each stage, the image of such a mapping is bounded into a certain region, and if the mapping is contractive (|C_\psi C_\phi| < 1), this can be applied to arbitrarily long sequence (and thus infinite horizon).

The paper is well-written and easy to follow. However, I have concerns about its novelty. Overall the paper seems to be a straightforward application of universal approximation theorem of deep neural network. The authors don’t specify when the condition will hold (e.g., |C_\psi C_\phi| < 1), what kind of empirical RNNs fall into this category and whether the trained models are operating in the contractive region.

Also with the same technique the authors also reach the same conclusion for nonlinear dynamics system (Sec. 4.2). From the proof, there is no much difference between linear case (Kalman Filter) and nonlinear case, since apparently DNN can fit everything. Then what are the new discoveries and takeaway messages? It is not clear what can we learn from this analysis and its impact is likely to be limited.

**Experience Assessment:**

I have published one or two papers in this area.

**Review Assessment: Checking Correctness Of Derivations And Theory:**

I assessed the sensibility of the derivations and theory.

**Review Assessment: Checking Correctness Of Experiments:**

N/A

**Review Assessment: Thoroughness In Paper Reading:**

I read the paper at least twice and used my best judgement in assessing the paper.

---

> ### Author Response · Authors · 2019-11-14
> **Response to Reviewer #3**
>
> Thank you for your detailed response.
>
> COMMENT: Overall the paper seems to be a straightforward application of universal approximation theorem of deep neural network.
>
> RESPONSE: Although the universal approximation theorem of deep neural network is the key ingredient in our analysis, the analyses of the accumulated error and asymptotic convergence are not so straightforward. In our paper, we first prove that the sufficient statistics and their approximations by DNN can be controlled in a bounded area (Lemma 2), and then give the accumulated error by iteration. To the best of our knowledge, our work is the first to derive an asymptotic result regarding RNN's expressive power in approximating stochastic system.
>
> COMMENT: The authors don’t specify when the condition will hold (e.g., $|C_\varphi C_\phi| < 1$),
>
> RESPONSE: Actually this condition can be satisfied by a general class of systems. We give a general class of systems that satisfies the condition $|C_\varphi C_\phi| < 1$ in Appendix A6 of the revised paper.
>
> COMMENT: what kind of empirical RNNs fall into this category and whether the trained models are operating in the contractive region.
>
> RESPONSE: Actually, in our paper, we consider RNN in a more general sense. We focus on the theoretical analysis rather than the empirical application, just as how Universal Function Approximation Theorem is derived. More specifically, we try to understand the boundary of what RNN can do. But still, investigating what kind of empirical RNNs fall into our category is an important and interesting research direction. As for the contractive region question, we actually do not understand what you actually mean. Do you mean that the outputs of the trained model just became smaller and smaller? $|C_\varphi C_\phi|<1$ does not necessarily imply the trained model is contractive. In our simulation (available upon requested), we also do not observe such "contractive" phenomenon.
>
> COMMENT: Also with the same technique the authors also reach the same conclusion for nonlinear dynamics system (Sec. 4.2). From the proof, there is no much difference between linear case (Kalman Filter) and nonlinear case, since apparently DNN can fit everything. Then what are the new discoveries and takeaway messages? It is not clear what can we learn from this analysis and its impact is likely to be limited.
>
> RESPONSE: Our idea about using sufficient statistics to capture the conditional distribution is inspired by the work of finite dimensional filters such as Benes filter. Kalman filter is a special case of finite dimensional filters and we use it to present our result so that it is more accessible to those without background knowledge of finite dimensional filter. There is no substantial difference between Sec. 4.1 and 4.2, since Sec. 4.1 is a special case of Sec. 4.2. Furthermore, our algorithm can be also used for general nonlinear filtering problems. Numerical experiments show that our algorithm can solve 10 dimensional nonlinear problem efficiently. When the length of sufficient statistics is infinite, we cannot use finite nodes in RNN to represent them, and this part of theoretic work is under study.

---

### Official Review · AnonReviewer1 · 2019-10-25
**Official Blind Review #1**

**Rating:** 3

**Review:**

The paper attempts to establish the asymptotic accuracy of the "RNN" (but not the RNN models that are well-known in the literature - see the below comments) as a universal functional approximator. It considers a general state space model and uses feedforward neural nets (FNN) to learn the filtering and forecast distributions. Based on the well-known universal approximation property of FNNs, the paper shows that their RNN-based filter can approximate arbitrarily well the optimal filter.

The paper targets an important problem in statistical learning and offers some interesting insights. However, their so-called RNN-based filter formulation is not anywhere close to the usual RNN models, such as Elman's basic RNN model or LSTM, that are currently known in the literature. Hence the paper's title "RNNs are universal filters" and the way it is presenting are confusing. I think what exactly the paper is about is as follows. They consider a general state space model, then use FNNs to approximate the so-called transition equation and observation equation of that SS model. Then, the resulting RNN-based filter is shown to be able to approximate well the optimal filter of the original SS model. I didn't check the proof carefully but I guess it's intuitively straightforward given the available results from the approximation capacity of FNNs.

Therefore, I suggest the authors re-write the paper carefully to reflect better the problem that it actually targets. The property of the RNN-based filter is interesting, but using it is, I believe, very difficult from a practical point of view. It's well-known that doing particle filters is computationally expensive, especially in high dimensions, and the RNN-based filter might have millions of parameters! Could the authors please give some comments/discussion about this issue?

Some minor points:
1)  what exactly does "synthesized" mean?
2) Page 5: "Note that the probability space ... with finite moment..." doesn't read well. What is the moment of probability space?




**Experience Assessment:**

I have read many papers in this area.

**Review Assessment: Checking Correctness Of Derivations And Theory:**

I assessed the sensibility of the derivations and theory.

**Review Assessment: Checking Correctness Of Experiments:**

N/A

**Review Assessment: Thoroughness In Paper Reading:**

I read the paper at least twice and used my best judgement in assessing the paper.

---

> ### Author Response · Authors · 2019-11-14
> **Response to Reviewer #1**
>
> Thank you for your detailed response.
>
> COMMENT: However, their so-called RNN-based filter formulation is not anywhere close to the usual RNN models, such as Elman's basic RNN model or LSTM, that are currently known in the literature. Hence the paper's title "RNNs are universal filters" and the way it is presenting are confusing. ...... Therefore, I suggest the authors re-write the paper carefully to reflect better the problem that it actually targets.
>
> RESPONSE: Thanks for your suggestion on making the title and the paper content more consistent. We totally understand your concern that our paper content may not reflect the title if RNN is interpreted as the usual RNN models known in the literature and used in practice. But in our paper, we use the term "RNN" in a more general and abstract sense. As mentioned in the book Deep Learning(by Ian Goodfellow, Yoshua Bengio and Aaron Courville), "Much as almost any function can be considered a feedforward neural network, essentially any function involving recurrence can be considered a recurrent neural network". In this sense, the RNN-based filter architecture can be viewed as RNN. And just as the derivation of classical Universal Function Approximation result was not based on a specific type of feedforward neural network such as ResNet, our derivation of Universal Filter Approximation result is not based on a specific type of RNN. But still, investigating our result's implications for the usual RNN models, such as Elman's basic RNN model or LSTM can be an interesting research direction.
>
> COMMENT: The property of the RNN-based filter is interesting, but using it is, I believe, very difficult from a practical point of view. It's well-known that doing particle filters is computationally expensive, especially in high dimensions, and the RNN-based filter might have millions of parameters! Could the authors please give some comments/discussion about this issue?
>
> RESPONSE: Actually, we have done the numerical simulation of this algorithm and it performs very well. We did not put the simulation part into the paper due to the limitation of the paper length. We have added a complete experiment section in Appendix A9 of the revised paper.
> The first example we consider is a 10 dimensional linear system. It is shown that just with the 90 hidden variables, RNN based filter can achieve the performance of Kalman filter. Besides, we also consider the cubic sensor problem, which is nonlinear. And the simulation results shows that our algorithm is very efficient, much faster than other state-of-the-art methods.
> Theoretically, in the particles filter (PF), the conditional density distribution is approximated by the empirical distribution of a cluster of particles. Therefore the PF is computationally expensive in high dimensional cases. However, we use the sufficient statistics to represent the conditional distribution in our algorithm, and apparently the number of the sufficient statistics can be much smaller than the number of particles in PF. For instance, we only need conditional mean and covariance in Gaussian linear case. And in Benes filter (Ref. Benes 1981), the conditional density of a class of nonlinear systems is given in terms of just 10 sufficient statistics.
> Besides the experiments in the paper, we also simulate a 10-dimensional cubic sensor problem, which is nonlinear. The result shows that our method achieves average mse 6.77 on each dimension and only takes 0.0009 seconds on a single server machine to compute the result on average. We will add this example to the paper if necessary.
>
> COMMENT: Some minor points: 1) what exactly does "synthesized" mean? 2) Page 5: "Note that the probability space ... with finite moment..." doesn't read well. What is the moment of probability space?
>
> RESPONSE: 1) Here, "synthesize" can be interpreted as "approximate to any accuracy desired". For example, "RNN can synthesize Kalman Filter" means "RNN can approximate Kalman Filter to any accuracy desired".
> 	2) This sentence has been changed into "We consider the probability space $(\Omega,\mathscr{F},\mathbb{P})$ with inner product $\langle x,y\rangle=\mathbb{E}[x^Ty]$ and norm $\|x\|:=\mathbb{E}^{1/2}[x^T x]$, which is a Hilbert space and denoted as $L^2(\Omega,\mathscr{F},\mathbb{P})$". Finite second moment means that $\|x\|<\infty$, where $x$ is a random variable.

---

### Official Review · AnonReviewer2 · 2019-10-25
**Official Blind Review #2**

**Rating:** 6

**Review:**

The paper shows that recurrent neural networks are universal approximators of optimal finite dimensional filters. More specifically, the paper provides a theoretical analysis on the approximation error for any stochastic dynamic system with noisy sequential observations.

I find the work interesting. Yet, I don't understand how it substantially deviates from previous work. The cited work by Schafer&Zimmermann only considers deterministic systems -- but what is the distinguishing idea of your paper that makes your more general analysis possible? To me it seems utilizing sufficient statistics? But working with sufficient statistics resembles lossly again working with a deterministic-like abstraction? While I understand that this a purely theoretical work, it would be instructive to have practical demonstrations, showing what's happening when learning is actually done. Finally, interpreting RNNs as bayesian filters was utilized in recent works, e.g. Recurrent Neural Filters.

As a side note, isn't eq.3 called the Chapman-Kolmogorov equation. Also, the book by Jazwinski is often cited -- I think if the main findings utilized in your text, e.g. the proof of Thm1, were replicated in the Appendix, your paper would feel more complete.

**Experience Assessment:**

I have published one or two papers in this area.

**Review Assessment: Checking Correctness Of Derivations And Theory:**

I did not assess the derivations or theory.

**Review Assessment: Checking Correctness Of Experiments:**

N/A

**Review Assessment: Thoroughness In Paper Reading:**

I read the paper at least twice and used my best judgement in assessing the paper.

---

> ### Author Response · Authors · 2019-11-14
> **Response to Reviewer #2**
>
> First of all, we'd like to thank you for the encouragement on our paper.
>
> COMMENT: The cited work by Schafer&Zimmermann only considers deterministic systems -- but what is the distinguishing idea of your paper that makes your more general analysis possible? To me it seems utilizing sufficient statistics? But working with sufficient statistics resembles lossly again working with a deterministic-like abstraction?
>
> RESPONSE: Our work substantially differentiates from the previous works in the following three aspects.
>    (1)We consider a stochastic system instead of deterministic system in the paper of Schafer\&Zimmermann. As you mentioned, one of the key steps in our theoretical developments is introducing Sufficient Statistics. But your understanding that after introducing the sufficient statistics, we can work with a deterministic-like problem is not totally true. For sure, introducing sufficient statistics simplifies our problem, allowing us to work in finite dimensional Euclidean space rather than infinite dimensional function space. But utilizing sufficient statistics does not make the system deterministic. As shown in equation (9) in our paper, $s_{k|k}=\varphi(s_{k|k-1}, y_k)$, where $y_k$ is the observation in step $k$. Since $y_k$ is random, the evolution of sufficient statistics can be viewed as a stochastic dynamic system.
> 	(2)	The distinguishing idea of our work is interpreting the hidden variable in recurrent neural networks as sufficient statistics, which can uniquely determine the conditional distribution of the state. And the evolution of hidden variables is interpreted as the evolution of statistics. Therefore the optimal filter can be approximated by recurrent neural networks for a class of systems, i.e. finite dimensional filter systems.
> 	(3)	In the work of Schafer\&Zimmermann and many others (like James Ting Ho Lo's work on neural filter), they did not consider the error in the iterative steps and the time horizon is finite. However, in our work, we analyze the iteration of the error delicately and prove that the estimation error can be sufficiently small as the time goes to infinity.
>
> COMMENT: While I understand that this a purely theoretical work, it would be instructive to have practical demonstrations, showing what's happening when learning is actually done.
>
> RESPONSE: Actually, we have done the numerical simulation of this algorithm and it performs very well. We did not put the simulation part into the paper due to the limitation of the paper length. We have added a complete experiment section in Appendix A9 of the revised paper.
>     The first example we consider is a 10 dimensional linear system. It is shown that just with the 90 hidden variables, RNN based filter can achieve the performance of Kalman filter. Besides, we also consider the cubic sensor problem, which is nonlinear. And the simulation results shows that our algorithm is very efficient, much faster than other state-of-the-art methods.
>
> COMMENT: Finally, interpreting RNNs as bayesian filters was utilized in recent works, e.g. Recurrent Neural Filters.
>
> RESPONSE: Thank you for your kindly remind and we did not know this 2019 new paper in arXiv when we submitted the paper. The results in the arXiv paper are similar to the June, 2018 graduation thesis of the first author of the current paper in the use of sufficient statistics, and emphasize on the design and implementation of the algorithm. However, our current paper focus on the error analysis and asymptotic convergence of the algorithm. And our work is inspired by the finite dimensional filter such as Benes filter.
> 	We have added this paper to our references' list.
>
> COMMENT: As a side note, isn't eq.3 called the Chapman-Kolmogorov equation. Also, the book by Jazwinski is often cited -- I think if the main findings utilized in your text, e.g. the proof of Thm1, were replicated in the Appendix, your paper would feel more complete.
>
> RESPONSE: Thanks a lot for your suggestions. 1) Yes, eq. 3 is Chapman-Kolmogorov equation and we have corrected it. 2) We have added the proof of Thm. 1 into the appendix. As for the proof of Lemma 1, it is long and requires extra definitions, so we do not list the proof here.

---

### Author Response · Authors · 2019-11-14
**About the New Version of the Paper**

We are grateful to the reviewers for their helpful and detailed comments on our manuscript. For the reviewers' convenience, the changes in the revision are in blue. The following are the major modifications in this revision:
1.	We add a complete proof of Theorem 1 in Appendix A1, as suggested by Reviewer #2.
2.	We give a general class of systems that satisfies the condition $|C_\varphi C_\phi| < 1$ in Appendix A6, as suggested by Reviewer #3.
3.	We add a complete experiment section in Appendix A9, as suggested by Reviewer #1 and Reviewer #2.

---

### Decision · Program_Chairs · 2019-12-19

**Decision:**

Reject

**Comment:**

Based on the Bayesian approach to filtering problem, the paper proves that RNN are universal approximators for the filtering problem.  Two reviewers, however, have doubts about the novelty and difficulty to get the result. Although I do not fully agree that Reviewer3 that the proof is just "DNN can fit anything" - it is not this case, but the concerns of Reviewer2 are more strong, especially about the usage of the term "recurrent neural network". The paper is purely theoretical and does not have any numerical experiments, which probably makes it too weak for ICLR in this form. However, I encourage the authors to continue to work on the subject, since the approach looks very interesting but it still very far from practice.